Interactions between donor Agreeableness and recipient characteristics in predicting charitable donation and positive social evaluation

Yarkoni Tal 1 tyarkoni@gmail.com
Ashar Yoni K. 2
Wager Tor D. 2
1 Department of Psychology, University of Texas at Austin , United States
2 Department of Psychology and Neuroscience, University of Colorado Boulder , Boulder, CO , United States
Cloninger C. Robert
Electronic publication date: 2015 Aug 18
Publication date: 2015
Volume: 3
Electronic Location ID: e1089
Received 2015 Mar 31; Accepted 2015 Jun 17
Copyright: © 2015 Yarkoni et al.
Copyright year: 2015
Copyright holder: Yarkoni et al.
License: This is an open access article distributed under the terms of the Creative Commons Attribution License, which permits unrestricted use, distribution, reproduction and adaptation in any medium and for any purpose provided that it is properly attributed. For attribution, the original author(s), title, publication source (PeerJ) and either DOI or URL of the article must be cited.
License URL: https://creativecommons.org/licenses/by/4.0/

Keywords: Personality, Charity, Agreeableness, Social evaluation, Charitable donation

Funding: John Templeton Foundation This project was supported by a John Templeton Foundation award to TDW. The funders had no role in study design, data collection and analysis, decision to publish, or preparation of the manuscript.

==============================
Agreeable people are more likely to display prosocial attitudes and helpful behavior in a broad range of situations. Here we show that this tendency interacts with the personal characteristics of interaction partners. In an online study (n = 284), participants were given the opportunity to report attitudes toward and make monetary donations to needy individuals who were described in dynamically generated biographies. Using a machine learning and multilevel modeling framework, we tested three potential explanations for the facilitatory influence of Agreeableness on charitable behavior. We find that Agreeableness preferentially increased donations and prosocial attitudes toward targets normatively rated as being more deserving. Our results advance understanding of person-by-situation interactions in the context of charitable behavior and prosocial attitudes.

Introduction

Human beings are a social species. Our desire and ability to cooperate with other humans in the face of potential costs to ourselves is arguably one of the chief reasons for our evolutionary success (Alexander, 1974; Axelrod & Hamilton, 1981). Yet the tendency to behave charitably is just that—a tendency. It is not universal, and each potential act of charity depends vitally on characteristics of both the helper and the receiver of help (Graziano et al., 2007). For example, highly Agreeable people—whose behavior is characterized by increased warmth, social affiliation, and compassion—are more likely to help others relative to low-Agreeable people (Carlo et al., 2005; LePine & Van Dyne, 2001). Analogously, some people are more likely to elicit help from others—e.g., if they are perceived as being highly likeable or less responsible for their misfortune (Appelbaum, 2002; Bekkers & Wiepking, 2011; Wagner & Wheeler, 1969). However, the influences of giver and recipient attributes on helping have most commonly been studied in isolation.

Here, we tested three alternative models of how Agreeableness may influence responding to needy others. First, highly Agreeable people may behave more charitably towards everyone, irrespective of recipient characteristics (Fig. 1A). Second, Agreeableness might manifest as a selective tendency to forgive bad behavior. That is, agreeable people might be more likely to help unfriendly social partners that less agreeable people would spurn, while being no less likely than disagreeable people to help out more pleasant social partners (Fig. 1B). Third, agreeable people might instead selectively reward positive behavior—that is, they might preferentially give help to targets who display prosocial traits, while being no more or less likely to help out targets who show less desirable traits (Fig. 1C). Establishing which of these models best captures the actual pattern of interaction between a giver’s Agreeableness level and a recipient’s characteristics can inform both theoretical models of personality and real-world policy choices regarding efforts to influence charitable behavior.

Figure 1 Three ways Agreeableness could hypothetically interact with recipient characteristics.

We assume that positively-evaluated recipients are more likely to elicit charitable behavior in all cases; however, this main effect could arise in different ways. In (A), High-Agreeable givers are more charitable than Low-Agreeable givers, and this effect is independent of partner characteristics. In (B), High-Agreeable givers preferentially overlook potential recipients’ negative behavior. In (C), High-Agreeable givers preferentially reward recipients who display more prosocial characteristics.

In the present study, we conducted a strong test of these competing models. In an online experiment, participants (n = 284) reported their attitudes toward dynamically generated biographies of needy individuals (i.e., “recipients”) designed to produce a wide range of evaluative impressions, and donated real money to real charities associated with those recipients. We used a machine learning and multi-level modeling framework to test for the presence of robust interactions between participants’ Agreeableness level and normative attitudes towards recipients. We further show that these effects are not explained by other major personality traits, and are primarily driven by compassion-related facets of Agreeableness rather than by acquiescence-related facets.

Methods

Participants

A nation-wide sample (n = 387) was recruited through Amazon’s Mechanical Turk service, 284 of who completed all parts of the study and provided usable data (183 women [64%]; 208 white [73%], 24 Asian or Pacific Islander [8%], 22 black [8%], 13 Hispanic [5%], 17 other or unknown [6%]; mean age = 33.5, range = 18–81, sd = 12.3). Previous studies have demonstrated that Mechanical Turk samples are more diverse than, and produce comparably reliable results to, traditional undergraduate samples (Buhrmester, Kwang & Gosling, 2011; Paolacci, Chandler & Ipeirotis, 2010). To maintain high data quality, we restricted participation (via MTurk’s built-in selection tools) to current US residents who had previously had at least 90 of their tasks approved. All participants read an informed consent sheet describing the study and all associated risks and benefits, and were required to click on a button that said “I Agree” before they were able to continue. Each participant was paid $1 for participation plus an additional $1 endowment to be kept or donated as desired. Participants who lacked variance in the outcome variable of interest for each analysis (i.e., who gave the same response on every trial) were excluded; thus the effective sample size for each analysis ranged from n = 200 to n = 284, depending on outcome measure (see below). Non-overlapping analyses of these participants’ data have been reported elsewhere (Y Ashar et al., 2015, unpublished data). All procedures were approved by the University of Colorado Boulder IRB (application #10-0210).

Personality measure

Personality was assessed using the Analog to Multiple Broadband Inventories (AMBI)—a 181-item public domain instrument that accurately recaptures scores on over 200 scales from 8 different broadband inventories—including the NEO-PI-R, HEXACO-PI , TCI-R, and several other widely used measures (Yarkoni, 2010). The AMBI was developed by applying a genetic algorithm to a large dataset drawn from the Eugene-Springfield community sample acquired by Goldberg and colleagues (Goldberg et al., 2006). Extensive convergent and discriminant validation analyses reported in Yarkoni (2010) demonstrated that the AMBI recaptures scores in 8 different broadband measures with extremely high fidelity despite the large reduction in length (a maximum of 5 items per scale). To minimize the number of comparisons performed in the present study, we focused exclusively on the AMBI version of the NEO-PI-R (Costa Jr& McCrae, 1992)—a widely used broadband inventory that provides scores for five major domains of personality (the so-called “Big Five” dimensions of Neuroticism, Extraversion, Openness, Agreeableness, and Conscientiousness) as well as 30 lower-order facet scales.

Procedure

Participants recruited through Mechanical Turk were directed to a custom online experiment written by the authors using the Ruby on Rails web framework. After providing consent and filling out basic demographic information (age, gender, and race), each participant was presented with 16 unique, realistic four-sentence biographies describing an individual in need along with a photograph and the name of an associated charity. We dynamically generated unique biographies in order to elicit a broader range of evaluative responses, relative to a small, static set of stimuli. Each biography was randomly constructed from a pool of 256 candidate sentences focused on one of 16 ‘hardships’ (e.g., suffering from cancer, having a substance abuse problem, etc.), and was randomly paired with one of 40 photographs and 24 names (photographs were paired with gender-consistent names). In total, over 4,000 unique randomly generated biography-photograph configurations were presented to participants during the course of this experiment, out of a total possible 280,000 biography-photograph configurations. A sample biography is: “Daniel has lung cancer. He only smoked cigarettes on weekends when he went out in high school and college. He attends church every Sunday. He has a reputation as somewhat of a gossip.” Additional samples are provided in Table S1.

Participants rated each of the 16 targets on 22 items (listed in Table S2) chosen to assess 7 different attitudes toward recipients, including: (1) feelings of tenderness, (2) personal distress over the other’s plight, (3) perceived neediness, (4) blaming the other for their suffering, (5) likeability of the other, (6) self-similarity to the target, 1 and (7) overall intent to help. These dimensions were chosen because of prior literature demonstrating their importance in prosocial responding to needy others (Batson et al., 2005; Batson, 2011; Eisenberg, Fabes & Spinrad, 2007; Greitemeyer & Rudolph, 2003; Vollhardt & Staub, 2011).

Participants additionally made a donation to each biography from their endowment (between $0 and $1, in 10-cent increments). To encourage participants to make independent choices on each trial, they were instructed that only one donation trial would be randomly selected, subtracted from their endowment, and donated.

Normative recipient ratings

Because every single biography presented to participants was dynamically generated and unique across the entire study, we generated normative attitudes and donation amounts for each biography using a predictive model. For each biography, we predicted ratings on the 6 attitudes (i.e., tenderness, distress, neediness, blame, likeability, intent to help) as well as charitable donation amounts from the explicitly manipulated properties of the stimuli (the 16 hardships, 256 sentences, 40 faces, and 24 names used to construct the biographies—representing 336 indicators in total). We used ridge regression—an L2-penalized variant of least-squares regression that is robust in the face of large numbers of correlated predictors.

The resulting estimates represent our best prediction as to the ratings and donation choices that the average participant would make when presented with each uniquely generated biography. Importantly, these estimates were based on the full dataset rather than on the research team’s subjective judgment or the results of a much smaller pilot study. In a five-fold cross-validated analysis (to mitigate overfitting) the model recaptured a substantial portion of the variance for all dimensions (Table S3).

Multilevel model

For each of the 7 outcome variables, we fit a multi-level model using the lme4 package for R (Baayen, Davidson & Bates, 2008) in order to quantify the relative contributions of normative attitudes/behavior toward recipients, participant Agreeableness, and their interaction. Formally, the model can be expressed as: Yts=γ00+γ01Agrees+γ10Normts+γ11Agrees∗Normts+u0s+u1sNormts+ets.

Here, Yts is the observed rating of recipient t made by subject s; Normts is the normative or expected rating for that recipient (produced by the ridge regression model described in the previous section); Agrees is Agreeableness score for subject s; and ets is the residual error. The fixed effects γ respectively model the grand intercept, the subject-level effect of Agreeableness, the trial-level effect of normative rating, and the cross-level interaction between normative rating and Agreeableness. The random terms, denoted by u, represent the subject-level intercepts and subject-specific effects of the normative ratings.

Results

Person-by-situation interaction

Table 1 summarizes the fixed-effects results from the multilevel model—that is, the main effects of normative attitudes or behavior towards recipients, of Agreeableness, and of their interaction—for each of the 6 attitudes plus donation amount. 2 For all recipient characteristics, Agreeableness consistently influenced both the main effect and interaction. Main effects were in the expected direction in all cases—i.e., highly Agreeable people were more likely, on average, to perceive recipients as more likeable, less responsible for their troubles, etc. To facilitate interpretation of the interaction between participant Agreeableness and normative attitudes toward recipients, Fig. 2 displays subject-level regression lines alongside summary lines for high- (+1 SD) and low (−1 SD)-agreeable subjects for the dimensions of perceived likeability and amount donated (for other dimensions, see Fig. S1). For all outcomes, Agreeable participants were disproportionately more charitable and prosocial toward targets normatively considered more deserving of donation and prosocial—supporting the hypothesis illustrated in Fig. 1C—rather than being more forgiving of normatively disliked targets, or indiscriminately being more charitable towards all recipients (cf. the competing hypotheses depicted in Figs. 1A and 1B).

Figure 2 Person-by-situation interaction in prediction of participants’ trial-by-trial behavior.

(A) likability ratings; (B) donation amounts. Colored lines reflect individual subject fits (grouped into discrete tertiles for visual clarity); black lines reflect the means for high-Agreeableness (>1 SD from mean; solid line) and low-Agreeableness (<1 SD from mean; dashed line) participants. For corresponding plots for other outcome variables, see Fig. S1.

Table 1 Results of multilevel model predicting behavior from recipient attributes and participant Agreeableness.

Behavioral variable	Recipient	A	Recipient:A	
Perceived responsibility	1.12 (0.03)***	−0.27 (0.04)***	0.06 (0.03).	
Perceived likeability	0.73 (0.03)***	0.20 (0.04)***	0.10 (0.03)***	
Perceived neediness	0.41 (0.02)***	0.20 (0.04)***	0.05 (0.02)*	
Felt sympathy	0.55 (0.02)***	0.23 (0.05)***	0.05 (0.02)*	
Felt distress	0.52 (0.02)***	0.18 (0.05)***	0.07 (0.02)**	
Intent to help	0.68 (0.03)***	0.38 (0.07)***	0.07 (0.03)*	
Donation amount	1.09 (0.06)***	0.40 (0.17)*	0.13 (0.06)*	
Notes.

* p < .05.

** p < .01.

*** p < .001.

. p < .1.

The ‘recipient’ and ‘A’ columns reflect the respective effects of normative target ratings and subject Agreeableness score on the behavioral ratings. Their interaction is captured in the ‘recipient:A’ column.

Comparison with other personality traits

Although our hypotheses pertained specifically to Agreeableness, for exploratory purposes, we repeated the above analyses for the other four Big Five domains (Extraversion, Neuroticism, Openness, and Conscientiousness). Results revealed systematic relationships between each trait and behavior (Table S3). For example, Extraversion was positively associated with mean ratings of tenderness and likeability, and Openness showed robust main effects similar to Agreeableness on most attitudes (i.e., highly Open people rated targets more favorably), but no interactions with normative attitudes. Interestingly, Conscientiousness interacted with recipient characteristics in a manner similar to Agreeableness, but showed no main effect (i.e., highly Conscientious people were no more likely to rate recipients as more needy or likeable on average, but their responses were generally more extreme at the low and high end than low-Conscientiousness participants). Importantly, including all other Big Five dimensions, as well as gender and age, as subject-level covariates in our multilevel models did not appreciably change any of the Agreeableness results reported above (Table S4).

Facet-level analysis of Agreeableness

Agreeableness is a broad dimension of personality that reflects a wide range of cooperative and affiliative behaviors; in the NEO-PI-R representation of personality, for example, Agreeableness comprises 6 narrower ‘facets’ that some researchers conceptualize as distinct aspects of Politeness and Compassion (DeYoung, Quilty & Peterson, 2007). We hypothesized that the trait and trait-by-situation effects we observed were likely to be driven primarily by the facets related to compassion for other people (Trust, Altruism, and Tender-Mindedness) rather than the facets associated with acquiescence and politeness (Straightforwardness, Compliance, and Modesty). Facet-level analyses provided modest support for this hypothesis: main effects and trait-by-situation interactions were somewhat more robust for the former facets—and particularly for Tender-Mindedness and Altruism—though they were present to some extent for the latter as well (Table S5).

Discussion

Previous studies have demonstrated that Agreeable people are more likely to display helping and charitable behavior and attitudes in a wide range of contexts (Carlo et al., 2005; Elshaug & Metzer, 2001; Graziano et al., 2007; Hilbig et al., 2013; Kamdar & Van Dyne, 2007; LePine & Van Dyne, 2001). The present study replicated these findings and further found that the influence of Agreeableness on social evaluation and charitable donation depended on characteristics of the donation targets. Agreeable people responded more positively to targets normatively perceived more positively, but showed little differential behavior towards targets normatively perceived less positively. Thus, highly Agreeable people do not appear to be indiscriminately more charitable and positive towards others (cf. Fig. 1A), nor does their increased charity specifically reflect a greater tendency to forgive negative behaviors (cf. Fig. 1C). Rather, they respond relatively more favorably to more deserving individuals (cf. Fig. 1B).

What explains the preferential influence of Agreeableness on evaluation of, and donation to, positively rated targets? One intuitive account is that highly Agreeable people may experience stronger feelings of compassion in response to—and are consequently more likely to help—potential recipients who have demonstrable prosocial characteristics. By contrast, it is arguably difficult for anyone, no matter how Agreeable, to feel much sympathy for a potential recipient described as, say, a drug addict who has no interest in quitting and will do anything to further his or her own interests. From this perspective, Agreeableness serves to modulate the gain on the amount of sympathy and compassion people experience in response to others’ need or suffering. Consistent with this account, the effects we observed were strongest for those facets of Agreeableness related to compassion and sympathy (i.e., Tender-Mindedness and Altruism) rather than to acquiescence or social desirability. Our findings are also consistent with a previous study (Graziano et al., 2007) in which participants high in Agreeableness produced relatively more favorable evaluations of minimally stigmatized or non-stigmatized social groups (e.g., blacks), but showed no difference in evaluation of highly stigmatized groups (e.g., child molesters).

Of course, this account is unlikely to be exhaustive. Charitable donation and social evaluation are complex phenomena, and we expect that multiple mechanisms contribute to their genesis and expression under different conditions. For example, given the long-standing debate over the extent to which self-reported Agreeableness reflects social desirability (Graziano & Tobin, 2002; Ones, Viswesvaran & Reiss, 1996), it is possible that the results we observe were driven in part by Agreeable participants’ desire to manage social impressions—i.e., that in a context where donations were made completely anonymously, we would observe dissimilar (or no) relationships between Agreeableness and charitable giving (DellaVigna, List & Malmendier, 2012; Izuma, Saito & Sadato, 2010). Moreover, the moderating influence of Agreeableness we observed is itself likely to depend on other factors that were unmeasured in the present study. For example, a recent study reported that highly Agreeable individuals may paradoxically evaluate others more negatively in cases where targets are described as behaving in overtly antisocial ways (Kammrath & Scholer, 2011). This latter account can be parsimoniously reconciled with our current findings if, for example, the effects we observed reflect an additive sum of two separate Agreeableness-mediated influences on behavior: first, a general positivity bias that applies across the spectrum of target behavior (cf. Fig. 1A); and second, a general increase in sensitivity to others’ social behavior, which applies at both ends of the spectrum (i.e., highly Agreeable people perceive prosocial behaviors relatively more positively and antisocial behaviors relatively more negatively).

More generally, our findings underscore the importance of considering trait and situational influences together rather than in isolation. While our findings broadly support previous studies demonstrating main effects of Agreeableness and recipient characteristics (Batson et al., 2005; Batson, 2012; Eisenberg, Fabes & Spinrad, 2007; Rudolph et al., 2004; Vollhardt & Staub, 2011) on donation, they also reveal more complex interactive effects that have potential implications for the understanding of personality and cooperative behavior. For example, if one construes Agreeableness solely in terms of its main effect on behavior—i.e., as a kind of general cooperation parameter that uniformly increases the likelihood of prosocial behavior in any given situation (cf. Fig. 1A)—then it follows that highly Agreeable people should also be more susceptible to exploitation when interacting with less scrupulous individuals. This problem has been explored extensively in game theoretic models and agent-based simulations, which demonstrate that highly-cooperative strategies cannot achieve stability in a population unless they are able to respond to defections by defecting in kind (Riolo, Cohen & Axelrod, 2001; Axelrod, 1981) Our results provide real-world behavioral support for such model-based insights by demonstrating that Agreeable people do indeed modulate their behavior in predictable ways when presented with targets who appear normatively undeserving of aid. Thus, understanding the dynamics governing specific personality traits may actually necessitate consideration of trait-by-situation interactions.

Lastly, our findings may have modest implications for the understanding and encouragement of charitable donation. We corroborated previous findings demonstrating that people are more likely to donate to recipients portrayed as being less blame-worthy (Greitemeyer & Rudolph, 2003) and in greater need (Batson, 2012), while also demonstrating that such appeals may have more pronounced effects on certain individuals—e.g., those of a highly Agreeable nature. Thus, in cases where organizations have insights into an audience’s characteristics, there may be utility in attempting to customize appeals for maximal effect—though it remains an open question as to how far the results reported here generalize to other real-world situations.

Supplemental Information

Supplemental Information 1 Supplementary tables and figures

Click here for additional data file.

Additional Information and Declarations

Competing Interests

Author Contributions

Human Ethics

Data Availability

1 We did not include self-similarity ratings in the present analyses, because these ratings (a) depended on the specific match between each target and each participant rather than on properties of the target alone, and (b) showed relatively little coherence across the 6 different similarity items.

2 Note that the strong effect of normative attitudes/behaviors on individual participants’ responses was entailed by our procedure, as these norms were indirectly extracted from the very same responses. We included these norms in the model reported in Table 1 strictly to afford a test of their interaction with participant Agreeableness.

The authors declare there are no competing interests.

Tal Yarkoni conceived and designed the experiments, performed the experiments, analyzed the data, contributed reagents/materials/analysis tools, wrote the paper, prepared figures and/or tables, reviewed drafts of the paper.

Yoni K. Ashar conceived and designed the experiments, performed the experiments, analyzed the data, contributed reagents/materials/analysis tools, wrote the paper, reviewed drafts of the paper.

Tor D. Wager conceived and designed the experiments, reviewed drafts of the paper.

The following information was supplied relating to ethical approvals (i.e., approving body and any reference numbers):

The study was reviewed and approved by the University of Colorado Boulder IRB (approval number 10-0210). Participants read a consent form online and were required to press a button that said “I AGREE” before they could proceed to the experiment.

The following information was supplied regarding the deposition of related data:

Data and analysis code has been deposited at: https://github.com/tyarkoni/YAW_PeerJ_2015.

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
