# Peer review of "Interactions between donor Agreeableness and recipient characteristics in predicting charitable donation and positive social evaluation"

_PeerJ, doi:10.7717/peerj.1089_

## Round 0.1 · original submission · Minor Revisions

Both reviewers found the paper to be well done and interesting but request that you extend your description of the sample and some other issues that may influence the generalizability of your findings.

Reviewer 1 ·

Basic reporting

No Comments

Experimental design

The use of Amazon's Mechanical Turk service is not described in enough detail that any evaluation can be made of it. Further, the Analog to Multiple Broadband Inventories is not explained in any detail. Since these are important measures for the study, better descriptions are needed.

Validity of the findings

No Comments

Additional comments

The sample used needs to be explained in greater detail. Are there any demographics about the sample? The sample itself may have inherent biases. What limitations does the sample pose for the study?

·

Basic reporting

This is an interesting study, linking self-reported personality traits to potentially relevant actions. The study is well executed. The research well designed to test the study hypotheses, and the execution of the study consistent and rigorous. The writing is clear, and the discussion follows from the results.

Experimental design

No comments

Validity of the findings

No comments

Additional comments

Some extra points the authors might consider including in their discussion:
1. Could the results be explained by generosity being driven by similarity? The recipients deemed more worthy by the highly agreeable participants were those made to seem more agreeable. This alternate explanation could be tested by seeing if highly conscientious participants preferentially rewarded more seemingly conscientious targets, more extrovert participants were more generous towards extrovert targets etc.
2. Is there an element of agreeableness that would be affected by the participant perceiving that his/her actions were being observed? Would the participants of the current study have made the same choices if they believed that people in their social environment would learn of their behavior? I raise this issue because it might have relevance to generalizability of the results.

---

## Round 0.2 · accepted · Accept

Your revisions have substantially improved the manuscript, which is a valuable contribution to the literature.